# Effective Capacitance from Equivalent Electrical Circuit as a Tool for Monitoring Non-Adherent Cell Suspensions at Low Frequencies

**DOI:** 10.3390/bioengineering9110697

**Published:** 2022-11-16

**Authors:** Alma De León-Hernández, Luisa Romero-Ornelas, Roberto G. Ramírez-Chavarría, Eva Ramón-Gallegos, Celia Sánchez-Pérez

**Affiliations:** 1Instituto de Ciencias Aplicadas y Tecnología, Universidad Nacional Autónoma de México, Ciudad de México 04510, Mexico; 2Escuela Nacional de Ciencias Biológicas, Instituto Politécnico Nacional, Ciudad de México 07738, Mexico; 3Instituto de Ingeniería, Universidad Nacional Autónoma de México, Ciudad de México 04510, Mexico

**Keywords:** effective capacitance, equivalent circuit, electrical double layer, impedance spectroscopy, non-adherent cell suspension

## Abstract

Analyzing the electrical double layer (EDL) in electrical impedance spectroscopy (EIS) measurement at low frequencies remains a challenging task for sensing purposes. In this work, we propose two approaches to deal with the EDL in measuring impedance for particles and non-adherent cells in an electrolytic suspension. The first approach is a simple procedure to compute a normalized electrical impedance spectrum named dispersed medium index (${\mathrm{DM}}_{i}$). The second is the EIS modeling through an equivalent electric circuit based on the so-called effective capacitance ($C_{\mathrm{ef}}$), which unifies the EDL phenomena. Firstly, as an experiment under controlled conditions, we examine polymer particles of 6, 15, and 48 μm in diameter suspended in a 0.9% sodium chloride solution. Subsequently, we used K-562 cells and leukocytes suspended in a culture medium (RPMI-1640 supplemented) for a biological assay. As the main result, the ${\mathrm{DM}}_{i}$ is a function of the particle concentration. In addition, it shows a tendency with the particle size; regardless, it is limited to a volume fraction of 0.03 × 10^−4^ to 58 × 10^−4^. The ${\mathrm{DM}}_{i}$ is not significantly different between K-562 cells and leukocytes for most concentrations. On the other hand, the $C_{\mathrm{ef}}$ exhibits high applicability to retrieve a function that describes the concentration for each particle size, the K-562 cells, and leukocytes. The $C_{\mathrm{ef}}$ also shows a tendency with the particle size without limitation within the range tested, and it allows distinction between the K-562 and leukocytes in the 25 cells/µL to 400 cells/µL range. We achieved a simple method for determining an $C_{\mathrm{ef}}$ by unifying the parameters of an equivalent electrical circuit from data obtained with a conventional potentiostat. This simple approach is affordable for characterizing the population of non-adherent cells suspended in a cell culture medium.

## 1. Introduction

Electrical impedance spectroscopy (EIS) is a powerful technique for characterizing suspensions of charged surfaces, e.g., particles and biological cells. The electrical double layer (EDL) is a phenomenon that occurs at low frequencies, and it appears on any electrically charged surface that interacts with a medium of free ions (electrolyte). The charged surface is developed for the particle when immersed in the electrolyte solution owing to the adsorption of ions onto the surface and/or ionization of dissociable groups on the surface [1,2,3]. In the presence of the EDL phenomenon, when an external electric field (E) is applied, the charge distribution surrounding the interface is distributed accordingly to the signal. As a result, when using an E at low frequencies, an electrical dipole moment is induced due to the EDL polarization [4], thereby causing dielectric dispersion, named α dispersion [5]. Since this occurs in cells and particles (dispersed medium) suspended in an electrolytic medium (liquid medium), these samples produce a similar α dispersion. Consequently, the particles suspended in an electrolyte develop a surface charge and can analog suspended cells in a biological medium by having an analog α dispersion [6].

The EDL also occurs as an undesirable effect known as electrode polarization (EP) that can overshadow the measured signal [7], and it obscures the bulk dielectric relaxation in the studying of dipolar motions in biological solutions [8]. In dielectric spectroscopy at low frequencies, the EDL of the electrodes (EEDL) is common to separate from the relaxation of the EDL of the sample. Compensating the EP effect by an algorithm is used to determine the sample’s dielectric constant [9] and the zeta potential [10]—another approach to reduce the EP is via the four-electrode technique implying high specific instrumentation [11]. Furthermore, relaxation times have been used to characterize particles in aqueous suspensions, establishing a relation between the particle size and concentration with the characteristic time [12,13,14]. One of the models used to describe the EDL is the Gouy–Chapman–Stern model, which establishes that an internal region (Stern layer) has firmly bound ions and an outer region (diffuse layer) with ions less associated, generating a disturbance in bulk properties [15]. Two capacitors in series model the EDL; therefore, the electrical impedance measurements have allowed studying the dependence between the signal potential, ions size, and temperature with the EEDL capacitance [16]. Furthermore, the EDL capacitance has been analyzed through the equivalent electrical circuit technique [17].

The EIS provides physiological information about the cell membrane and the intra- and extracellular medium [18]. From mHz to a few kHz, there is α dispersion associated with ionic species diffusion processes related to the cell membrane potential and the displacement of surrounding counterions. In the kHz and up to tens of MHz range, β dispersion occurs generated by the membrane polarization due to the charge distribution between the intracellular and extracellular medium, creating an electric dipole. Finally, the γ region between MHz and GHz is associated with water molecules and some proteins [19]. From α dispersion, it is possible to determine parameters associated with the physicochemical characteristics of the cell. When the cell is excited by an electrical field at low frequencies, its membrane behaves like a capacitor [20,21]. Hence, the electrical current does not travel through the intracellular medium and surrounds the cell, giving information about its shape and size [22]. The α dispersion significantly influences cell suspension in a culture medium: an EDL is formed around every cell since free ions of opposite charge are available, having a distribution of ions in the membrane and one in the surrounding region. The membrane is a negatively charged surface for almost all cells due to the predominance of negatively charged groups such as carboxylates and phosphates [23]. Consequently, the membrane attracts positive charges, resulting in the formation of the EDL.

The evaluation of dynamic changes induced by cytotoxic agents can be studied by electrical characterization [24,25]. Typically, such assays are performed through an electrochemical technique known as electrical cell-substrate impedance sensing (ECIS) [26], which implies cells grow as adherent monolayers at the surface of the electrodes and real-time measurement. In general, ECIS cytotoxicity assays are carried out by monitoring the cell detachment from the surface of the electrodes through a parameter called cell index (CI) [27]. The CI normalizes the cell monolayer electrical impedance with the culture medium measurement. The methods based on cell attaching and spreading out on the electrode have more significant limitations in non-adherent cells, such as blood cells, cancer cells, stem cells, etc. Monitoring non-adherent cells by techniques such as ECIS requires the use of substrates made with complex materials and procedures, such as graphene oxide [28], carbon nanotubes [29], or magnetic nanoparticles [30]. Another disadvantage is the evaluation under inadequate physiological conditions since dynamic interactions are not allowed, and interactions mediated by ions, proteins, and phospholipids, among others, are modified [31].

Our previous work showed EIS characterization of particle suspensions in a saline buffer in the spectral band of 1 kHz to 1 MHz for the excitation signal (β dispersion). We analyzed the spectra using an equivalent electrical circuit based on the Randles configuration [32]. The findings showed changes in the electrical parameters depending on the particle concentration [33], thereby establishing an attractive way to evaluate dispersed particles in an electrolyte; nevertheless, we did not examine the EDL phenomena. To go further, in the present work, we introduce the characterization by EIS in α dispersion, which remains a challenging task in assessing interfaces at low frequency. To show the performance of the proposal, we test suspensions of Poly (methyl-methacrylate) (PMMA) particles in a commercial saline solution (physiological serum), with the cell line K-562 (chronic myeloid leukemia cells) and leukocytes, both non-adherent cells, suspended in RPMI-1640 culture medium. The spectral range of measurements is 10–1000 Hz, where the EDL phenomenon is predominant, and, therefore, multiple EDLs are formed in the dispersed medium modifying the EEDL and bulk properties. We analyze EIS data with two methods; the first is directly computed data, determining a normalized impedance magnitude to observe only the electrical response of the dispersed medium. The second way is a parametric analysis, using the so-called effective capacitance ($C_{\mathrm{ef}}$), which is analytically derived from the elements of a proposed electrical circuit. The advantages and disadvantages of both methods are shown. Finally, we used the $C_{\mathrm{ef}}$ as a unified parameter for estimating the concentration and size of the dispersed medium, such as micrometric particles in a mimetic experiment or non-adherent biological cells. Contrary to the conventional methods, we show that EEDL could be exploited to extract valuable information from the dispersed medium, using a parameter that unifies several electrical data determined with a common potentiostat. In summary, the main contribution of this work is to provide an attractive and affordable method for characterizing, at low frequencies, non-adherent cells suspended in a cell culture medium. This method allows estimating the cell concentration, which could be helpful in several applications, such as monitoring cell growth, mobility, or death in biological assays.

## 2. Theoretical Framework

The Gouy–Chapman–Stern (GCS) model states that the EDL ion distribution is divided into two regions, the compact layer (Stern layer) and the diffuse layer that extends to the bulk solution (Figure 1). The inner Helmholtz plane (IHP), given by the water molecules, and the outer Helmholtz plane (OHP), drawn from the center of the closest solvated ions, form the Stern layer. This layer does not depend on the potential at the charged surface, while the diffuse layer varies as a potential function [17].

According to the GCS model, the compact layer and the diffuse layer can be interpreted as two capacitors in series:(1)$$\frac{1}{C_{\mathrm{DL}}}=\frac{1}{C_{S}}+\frac{1}{C_{\mathrm{Dif}}},$$
where $C_{\mathrm{DL}}$ is the EDL capacitance, $C_{S}$ is the Stern layer capacitance, and $C_{\mathrm{Dif}}$ is the diffuse layer capacitance.

The EDL capacitance is given by [34]:(2)$$\frac{1}{C_{\mathrm{DL}}}=\frac{x_{\mathrm{OHP}}}{{\varepsilon \varepsilon }_{0}}+\frac{1}{{\left(\frac{2{\varepsilon \varepsilon }_{0}z^{2}e^{2}n^{0}}{k_{B}T}\right)}^{\frac{1}{2}}\cosh\left(\frac{\mathrm{ze}\psi }{2k_{B}T}\right)},$$
where $x_{\mathrm{OHP}}$ is the distance between the charged surface and the OHP, ε is the relative permittivity of the medium, $\varepsilon_{0}$ is the permittivity of vacuum, $n^{0}$ is the ionic concentration, z is the valence of the ions, e is the electron charge, k_B_ is Boltzmann’s constant, T is the absolute temperature, and ψ is the potential at *x*_OHP_ with respect to the bulk solution. This model predicts that as the surface has a higher charge, the diffuse layer will be more compact, and the total capacitance will increase, assuming ε is constant along *x*. Treating ε as a constant may be incorrect since its value in the Stern and diffuse layers may differ. The dipoles in the Stern layer are highly aligned with the electric field, and, thus, ε may have a strong decrease [35]. In the case of the diffuse layer, the dipoles have a less aligned arrangement with respect to the electric field, so ε has a smaller decrease.

The EDL may not have an ideal capacitor behavior because the charged surface may be porous, rough, and heterogeneous, resulting in a surface dispersion of the EDL around the dispersed medium [13]. An electrical element called a constant phase element (CPE) is usually used to model this non-ideal behavior [36,37]. The electrical impedance of a CPE is defined as:(3)$$Z_{\mathrm{CPE}}=\frac{1}{T{\left(j\omega \right)}^{P}}=\frac{1}{T{\left(\omega \right)}^{P}}\left(\cos\left(\frac{\pi P}{2}\right)-j\sin\left(\frac{\pi P}{2}\right)\right),$$
where T [Fs^P−1^] is the CPE constant, $j=\sqrt{-1}$ is the imaginary number, ω is the angular frequency, and the variable P is 0≤P≤1. The case where P=1 describes an ideal capacitor, while the case where P=0 describes an ideal resistor [13]. The proposed equivalent electrical circuit to model the electrical impedance spectra for particle or cell suspensions is shown in Figure 2a. The ${\mathrm{CPE}}_{s}$ represents the properties of the bulk suspension, and $C_{e}$ represents the capacitance associated with the electrolyte-electrode interface. Finally, $R_{d}$ and ${\mathrm{CPE}}_{d}$ represent the resistance and capacitance of the multiple EDLs around the dispersed medium. Several methods allow the CPE impedance to be associated with a capacitance value by unifying the CPE constant and exponent. This capacitance is so-called effective capacitance [15,38,39,40,41]. 

In the proposed circuit (Figure 2a), the effective capacitance associated with the dispersed medium ($C_{d}$) is in parallel with the capacitance of the electrodes $C_{e}$, resulting in a total effective capacitance ($C_{\mathrm{ef}}$) given by:(4)$$C_{\mathrm{ef}}=C_{e}+C_{d}\;$$

The capacitance $C_{d}$ is determined using an analogy with the Cole-Cole model [42,43], resulting $C_{\mathrm{ef}}$ as:(5)$$C_{\mathrm{ef}}=C_{e}+{\left(T_{d}{*R}_{d}^{1-P_{d}}\right)}^{\frac{1}{P_{d}}}\;$$
where $T_{d}$ and $P_{d}$ are the constant and exponent of the ${\mathrm{CPE}}_{d}$, respectively.

## 3. Materials and Methods

### 3.1. PMMA Particle Suspensions (Biological Phantom) Preparation

PMMA particles (p) of different diameter (∅) were used to make three sets of suspensions (s_∅1,_ s_∅2,_ s_∅3_) in physiological serum (PS). We considered three particles size of ∅_1_ = 6 μm, ∅_2_ = 15 μm, and ∅_3_ = 48 μm. These sizes were chosen to encompass a similar range to those reported for leukocytes, between 6 and 20 μm in diameter [44,45], and for K-562 cells, between 12 and 28 μm in diameter [46,47,48]. The PS is a NaCl solution with a concentration of 0.9% corresponding to the molarity of 0.154 mol/L. Serial dilutions were made starting from an initial suspension of 400 p/μL, resulting in five concentrations (c_i_ for i = 1,2,3,4,5) of 25, 50, 100, 200, and 400 p/μL for each particle size. The suspensions were stabilized with 0.5% sodium dodecyl sulfate to avoid agglomeration of the particles.

### 3.2. Non-Adherent Cell Suspensions Preparation

Human cell suspensions were made with non-adherent cells, the first with a cancer cell line and the second with leukocytes. The cancer cell line was chronic myeloid leukemia K-562 (ATCC^®^ CRL-1593.2) that was cultured in RPMI-1640 culture medium supplemented with 10% fetal bovine serum (FBS) and 1% penicillin-streptomycin at 37 °C and 5% CO_2_. It was propagated when reaching a confluence of approximately 85%, renewing the medium twice a week. Leukocytes were obtained from a human peripheral blood sample. Erythrocytes were removed using a lysis buffer (15.5 mM NH_4_Cl, 1 mM KHCO_3_, and 0.01 mM EDTA). We prepared a mixture of 200 µL of blood and 2 mL of lysis buffer. After 5 min, the mixture was centrifuged at 300× *g* for 10 min and resuspended in PBS 1X, doing this twice to remove the lysis solution. The separated leukocytes were suspended in RPMI-1640 medium with 10% decomplemented FBS and 1% penicillin-streptomycin and left for 24 h at 37 °C and 5% CO_2_.

Cell suspensions were made in RPMI-1640 supplemented with 10% FBS and 1% penicillin-streptomycin for EIS measurements. Following the methodology of half-fold serial dilution, starting from an initial suspension of 400 cells/μL, we obtained five concentrations ($c_{\mathrm{cell}}{}_{i}$) for each non-adherent cell. The cell suspensions were maintained at 37 °C in Eppendorf tubes.

### 3.3. Numerical Simulations

The analysis of the $C_{\mathrm{ef}}$ electrical behavior was carried out through the equivalent electrical circuit numerical simulation. We set up values of the electrical elements (see Table 1), $R_{d}$ and $T_{d}$ have fixed values, and we test three combinations for values of $C_{e}$ and $P_{d}$, being the variables that cause a significant change in $C_{\mathrm{ef}}$ involved with the EEDL and the dispersed medium EDL. The electrical impedance spectra for the different cases are represented as Bode and Nyquist diagrams (see the Section 4).

### 3.4. EIS Measurements and Analysis

EIS measurements were made with a commercial potentiostat (PalmSens4) in a 10–1000 Hz frequency range, with an excitation signal of 0.05 [V]. We used DropSens™ DRP-G-IDE555 electrodes embedded in a poly (dimethylsiloxane) (PDMS) chamber made by soft molding with an area of 35 mm^2^ and a height of 3 mm (Figure 2b). The samples were thermalized to 37 °C. The chamber was filled with 100 μL of the sample, previously re-suspended, and the impedance spectrum was obtained immediately, taking about 1 min. We cleaned the chamber with the PS solution at the end of each measurement. Three aliquots for each concentration were measured, reporting the average impedance spectra. We started with the liquid medium, followed by the suspension with i = 1 until we finished with i = 5. For EIS analysis; the impedance magnitude spectrum is normalized according to [49,50]:(6)$${\mathrm{DM}}_{i}=\frac{{\left|Z\right|}_{S}-{\left|Z\right|}_{\mathrm{LM}}}{{\left|Z\right|}_{\mathrm{LM}}}$$
where ${\mathrm{DM}}_{i}$ is the dispersed medium index, ${\left|Z\right|}_{S}$ and ${\left|Z\right|}_{\mathrm{LM}}$ are the impedance magnitude of the suspension and the liquid medium, respectively.

We fitted the experimental EIS data with the proposed electrical circuit (Figure 2) using the Levenberg–Marquardt optimization algorithm (Table 2), and through its electrical parameters, we calculated the $C_{\mathrm{ef}}$. Finally, we analyzed the ${\mathrm{DM}}_{i}$ and $C_{\mathrm{ef}}$ behavior in function of the suspension concentration.

## 4. Results and Discussion

### 4.1. Numerical Simulations

The data for the five cases simulated are depicted in Table 1. In cases a, b, and c, $C_{e}$ is a constant, and $P_{d}$ value is 0.7, 0.8, and 0.9, respectively, implying the increase in the dispersed medium capacitance and, therefore, the increase in $C_{\mathrm{ef}}$. In cases c, d, and e, $P_{d}$ is a constant and $C_{e}$ value is 1, 2, and 4 [μF], respectively. These three last cases imply that an increase in the electrode capacitance, either due to a change in electrolyte ionic concentration or specific adsorption [42], causes an increase in $C_{\mathrm{ef}}$. 

Figure 3a,b shows the impedance spectra for cases in Table 1 from 10 to 1000 Hz, for magnitude (|Z|) and phase angle (θ), respectively. For cases, a, b, and c, an increase in $P_{d}$ causes an increase in $C_{d}$. Below 10^2^ Hz, the increase in $C_{d}$ causes a decrease in impedance magnitude. Regarding the phase, there is an increase in the angle visualized as a crest whose maximum value increases with $C_{d}$. For cases c, d, and e, $C_{e}$ increases cause a decrease in both the magnitude and the phase angle. The maximum value of the peak in the phase angle decreases when $C_{e}$ increases. Another graphical representation of an impedance spectrum is a Nyquist plot; each point represents the magnitude, and phase corresponding to a particular frequency, consequently, is a more compact representation. The diagram is in the complex plane, having the negative of the imaginary part versus the real part, considering the frequency as an implicit variable. In Figure 3c, the Nyquist diagrams for cases depicted in Table 1 are represented. For cases a, b, and c, on the right side (lower frequencies), an inclined spike changes when $C_{d}$ increases, which is associated with the species diffusion from the bulk solution to the interface electrode–electrolyte [32]. There is an increase in the real part (Re(Z)) and a decrease in the imaginary part (Im(Z)). On the left side (higher frequencies), there is a depressed semicircle that describes the kinetic of the species close to the electrode–electrolyte. Thus, an increase in the capacitance associated with the dispersed medium causes an ionic redistribution synthesized as an increase in the $C_{\mathrm{ef}}$ parameter. For cases c, d, and e, on the right side, a shortening of the inclined spike is observed when $C_{e}$ increases. There is a decrease in both the real and imaginary parts. On the left side, the tendency to form a semicircle prevails; if $C_{e}$ increases, the size of the semicircle increases. Therefore, an increase in the capacitance associated with the electrode is also synthesized as an increase in the parameter $C_{\mathrm{ef}}$. From these simulations, we can observe that diverse behaviors in the impedance diagrams, which imply changes in the capacitive effects, can be summarized through the effective capacitance extracted from unifying the electrical parameters of the equivalent circuit using Equation (5).

### 4.2. EIS Analysis for PMMA Particle Suspensions (Biological Phantom)

The Nyquist plots for s_∅1_ (Figure 4a) show a decrease in the real and imaginary part of the impedance as the number of particles increases. For s_∅2_ (Figure 4b), impedance is decreased between c_1_ and c_2_, and for c_3_, c_4_, and c_5_, the change is minor. Regarding s_∅3_ (Figure 4c), no significant changes are observed for all the suspensions, so to notice changes directly from the Nyquist diagrams is quite challenging.

Figure 4d depicts the ${\mathrm{DM}}_{i}$ spectra for particles of $\emptyset_{1}$. ${\mathrm{DM}}_{i}$ basal value is zero, which corresponds to the physiological serum (PS). As can be seen, the ${\mathrm{DM}}_{i}$ value increases as the number of particles increases, following a consecutive order for the five concentrations. For particles of $\emptyset_{2}$ (Figure 4e), ${\mathrm{DM}}_{i}$ is increased for the five concentrations; however, there is a minor change between c_3_, c_4_, and c_5_. Finally, for particles of $\emptyset_{3}$ (Figure 4f), there is an increase in ${\mathrm{DM}}_{i}$ value; nevertheless, it is a small change for the five concentrations relative to each other.

A slight increase in ${\mathrm{DM}}_{i}$, when the particle is larger, can be explained by the difference in the volume fraction (V_f_) of the suspensions. The suspension c_1_ has 2500 particles and a V_f_ × 10^−4^ of 0.03, 0.4, and 14 for particles of $\emptyset_{1}$, $\emptyset_{2}$, and $\emptyset_{3}$, respectively. Therefore, the dispersed medium volume increases with the particle size (Table 3). For particles of $\emptyset_{1}$ (Figure 4d), the ${\mathrm{DM}}_{i}$ spectrum for c_1_ is close to the PS, indicating it is the most part liquid medium. Between c_1_ and c_5_ there is an increase from 0.02 to 0.19, at 50 Hz (the frequency in which ${\mathrm{DM}}_{i}$ has the most significant differences), for a V_f_ × 10^−4^ of 0.03 to 0.5. For particles of $\emptyset_{2}$ (Figure 4e), the increment in the ${\mathrm{DM}}_{i}$ value persists as the number of particles increases, going from 0.14 to 0.28, at 50 Hz, between c_1_ and c_5_. Compared to the particles of $\emptyset_{1}$, there is a more significant difference concerning the PS medium. However, the samples c_3_, c_4_, and c_5_ have similar values, indicating that the sensitivity decreases after a V_f_ =2 × 10^−4^. Finally, for particles of $\emptyset_{3}$, the ${\mathrm{DM}}_{i}$ increases from 0.22 to 0.25, at 50 Hz, between c_1_ and c_5_. Between c_1_ and PS, there is a more significant difference compared to the particles of $\emptyset_{1}$ and $\emptyset_{2}$. However, the differences for c_1_, c_2_, c_3_, c_4_, and c_5_ are not significant because they are samples with a larger V_f_, ranging from 14 × 10^−4^ to 230 × 10^−4^.

Figure 5a shows the ${\mathrm{DM}}_{i}$ as a function of the particle concentration (c_i_) at 50 Hz. The goodness of the fit is R^2^ = 0.98, R^2^ = 0.89, and R^2^ = 0.94 with a sensitivity of 0.15, 0.12, and 0.03 μL/p for $\emptyset_{1}$, $\emptyset_{2}$, and $\emptyset_{3}$, respectively. Nevertheless, for the most concentrated suspensions c_4_ and c_5_, ${\mathrm{DM}}_{i}$ has no consecutive values for the particle size since the plot for $\emptyset_{3}$ is between the plots for $\emptyset_{1}$ and $\emptyset_{2}$.

Figure 5b shows the $C_{\mathrm{ef}}$ as a function of the particle concentration, the goodness of the fit is R^2^ = 0.97, R^2^ = 0.95, and R^2^= 0.98 with a sensitivity of 1.7 × 10^−7^, 0.8 × 10^−7^, and 0.5 × 10^−7^ μL × F/p for $\emptyset_{1}$, $\emptyset_{2}$, and $\emptyset_{3}$, respectively. The $C_{\mathrm{ef}}$ has a higher value when there are more suspended particles, implying the effective capacitance of the EDLs is greater due to the increase in the net charged surface. According to the results, for a frequency of 50 Hz, ${\mathrm{DM}}_{i}$ is a function of the particle concentration for $\emptyset_{1}$, $\emptyset_{2}$, and $\emptyset_{3}$. Likewise, it is possible to recognize the particle size for more diluted suspensions, showing limitations for more concentrated suspensions with a large V_f_. On the other hand, $C_{\mathrm{ef}}$ allows a concentration-dependent fitting curve, and we can recognize the three particle sizes for the five suspensions. In such a way, impedance spectra analysis by the unified parameter $C_{\mathrm{ef}}$ presents the advantage concerning the ${\mathrm{DM}}_{i}$ of monitoring particle concentration and size for suspensions with a large V_f_.

### 4.3. EIS Analysis for Non-Adherent Cell Suspensions

Applying the same methodology described above, we obtained the ${\mathrm{DM}}_{i}$ spectra for cell suspensions in RPMI-1640 supplemented, both for the K-562 cells (Figure 6a) and leukocytes (Figure 6b). Figure 6a shows a progressive increase for the five concentrations, with a ${\mathrm{DM}}_{i}$ higher value as there are more cells suspended. The maximum standard deviation (σ) for K-562 cell suspensions was 3%. Regarding leukocyte suspensions, $c_{\mathrm{cell}}{}_{i}$ for i = 1,2,3,4,5, the maximum standard deviation was 36%, 16%, 3%, 6%, and 1%, respectively (Figure 6b). Notice that σ increases while the leukocyte number decreases, which could correspond to a concentration variation between aliquots due to a very low V_f_. Figure 7a shows ${\mathrm{DM}}_{i}$ as a function of cell concentration at 50 Hz, having an R^2^ = 0.98, and R^2^ = 0.77 with a sensitivity of 0.06, and 0.19 μL/cells for K-562 cells and leukocytes, respectively. ${\mathrm{DM}}_{i}$ value is different between K-562 cells and leukocytes only for the most diluted suspension ($c_{\mathrm{cell}}{}_{1}$).

Through the electrical circuit elements, we estimated the $C_{\mathrm{ef}}$ for the cell suspensions (Figure 7b). The goodness of the fit for K-562 cells is R^2^ = 0.95 with a sensitivity of 0.07 μL × F/cells, showing a clear relationship with cell concentration. For leukocytes, R^2^ = 0.92 with a sensitivity of 0.28 μL × F/cells. The $C_{\mathrm{ef}}$ function for K-562 cells and leukocytes, in contrast with ${\mathrm{DM}}_{i}$, are curves that do not intersect. Therefore, with the unified parameter $C_{\mathrm{ef}}$, we can analyze the impedance spectrum depending on cell concentration and type.

## 5. Conclusions

This work could characterize the EIS of a dielectric medium suspended in an electrolytic solution through a curve fitting depending on concentration. Two approaches were used for the EIS analysis. The first was to normalize the impedance magnitude with the medium liquid measurement named dispersed medium index (${\mathrm{DM}}_{i}$). The second approach was to calculate the unified parameter named effective capacitance ($C_{\mathrm{ef}}$), which synthesizes the electrical double layer of the electrode and the disperse medium by fitting an equivalent circuit with Levenberg–Marquardt method. The main findings are:The normalized impedance is a function of particle concentration and size for diameters of 6 and 15 μm, showing limitations for particles of 48 μm, starting with a volume fraction of 58 × 10^−4^. The sensitivity of the curve decreases with the particle size;The effective capacitance is a function of particle concentration and size for diameters of 6, 15, and 48 μm, evaluated in a volume fraction of 0.03 × 10^−4^ to 230 × 10^−4^. The sensitivity of the curve also decreases with the particle size;For non-adherent cell suspensions, the normalized impedance is not significantly different for K-562 cells and leukocytes. In contrast, the effective capacitance has a well distinguishable curve depending on concentration for each cell type, evaluated in a range of 25 cells/μL to 400 cells/μL with 100 μL of volume sample;The normalized impedance is a simple approach that only requires arithmetic treatment of the data, having limitations for analyzing changes in the sample size, which could be a drawback for a biological assay. On the other hand, effective capacitance is a more robust approach, which requires an optimization algorithm to determine the values of electrical circuit components. Nonetheless, it shows a better result for the sample size analysis.

According to the EIS results, the electrode polarization influence is predominant. Nevertheless, when we joined the phenomena present at low frequencies through the effective capacitance determined by unifying the electrical parameters of an equivalent circuit, the variations exhibited depend on the dispersed medium concentration and have a linear behavior. In this way, the effective capacitance applied to non-adherent cell suspensions could be a tool for monitoring changes in the population in proliferation or cell death assays. 

## Figures and Tables

**Figure 1 bioengineering-09-00697-f001:**
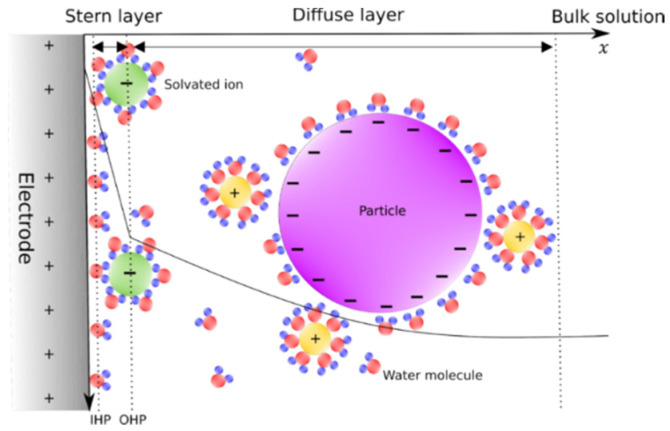
EDL on a charge surface according to the Gouy–Chapman–Stern model.

**Figure 2 bioengineering-09-00697-f002:**
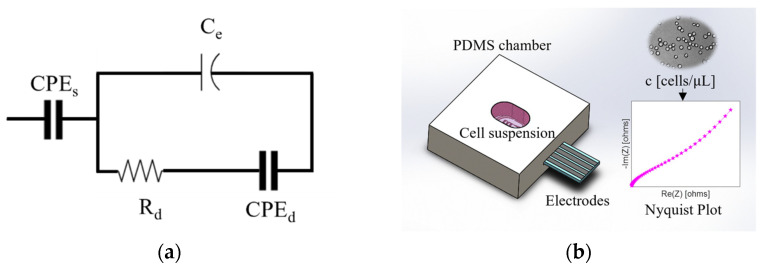
(**a**) Equivalent electrical circuit to model the electrical impedance data of particle suspensions or biological cell suspensions; (**b**) electrodes embedded in a PDMS chamber for EIS measurements.

**Figure 3 bioengineering-09-00697-f003:**
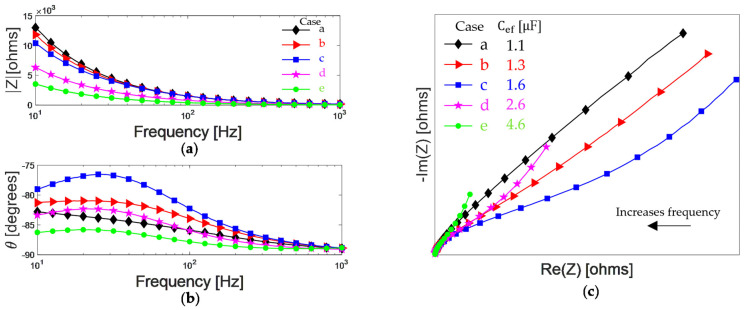
Impedance magnitude (**a**), phase angle (**b**), and Nyquist plot (**c**) of the proposed electrical circuit for cases depicted in Table 1.

**Figure 4 bioengineering-09-00697-f004:**
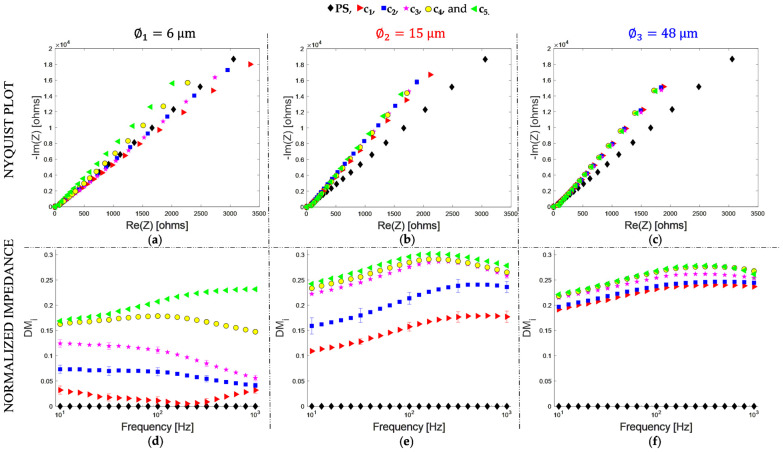
The electrical impedance spectrum of PMMA particle suspensions in PS is represented as Nyquist Plot (**a**–**c**) and as ${\mathrm{DM}}_{i}$ (**d**–**f**) for $\emptyset_{1}\;$ (**a**,**d**), $\emptyset_{2}\;$ (**b**,**e**), and $\emptyset_{3}$ (**c**,**f**). Concentrations are represented by the symbols.

**Figure 5 bioengineering-09-00697-f005:**
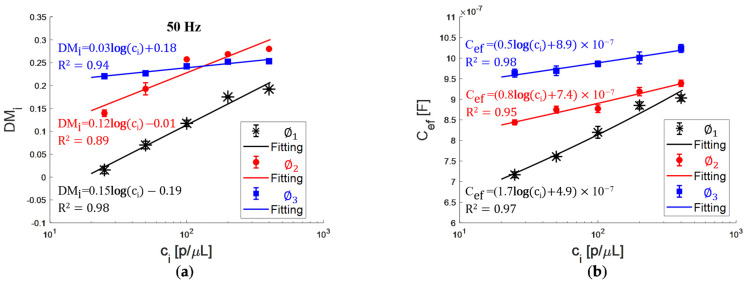
(**a**) ${\mathrm{DM}}_{i}$ at 50 Hz and (**b**) $C_{\mathrm{ef}}$ as a function of PMMA particle concentration for $\emptyset_{1}$, $\emptyset_{2}$, and $\emptyset_{3}$ (symbols-data and solid lines-fittings).

**Figure 6 bioengineering-09-00697-f006:**
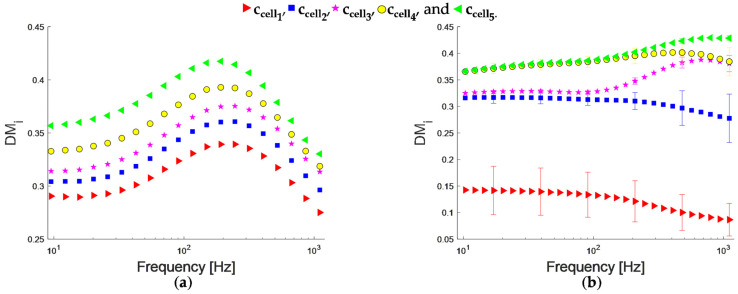
${\mathrm{DM}}_{i}$ of (**a**) K-562 cells and (**b**) leukocytes suspended in RPMI-1640 supplemented. Concentrations are represented by the symbols.

**Figure 7 bioengineering-09-00697-f007:**
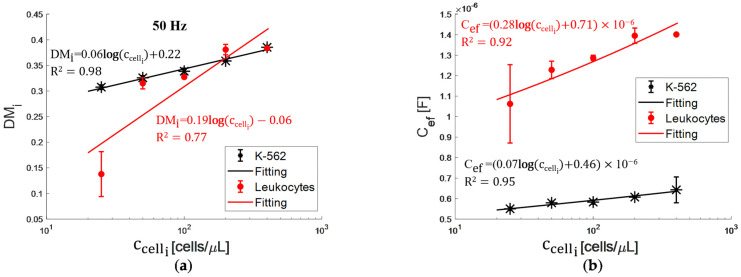
(**a**) ${\mathrm{DM}}_{i}$ at 50 Hz and (**b**) $C_{\mathrm{ef}}$ as a function of cell concentration (symbols-data and solid lines-fittings).

**Table 1 bioengineering-09-00697-t001:** Electrical values for the equivalent circuit (Figure 2a) with $T_{s}=0.1{\;\mathrm{Fs}}^{P-1},{\;P}_{s}=0.3$, $R_{d}=10\;k\Omega$, $T_{d}=1\times 10^{-6}{\;\mathrm{Fs}}^{P-1}$ and the retrieved effective capacitance $C_{\mathrm{ef}}$.

Case	$C_{e}$ [μF]	$P_{d}$	$C_{ef}$ [μF]
a	1	0.7	1.1
b	1	0.8	1.3
c	1	0.9	1.6
d	2	0.9	2.6
e	4	0.9	4.6

**Table 2 bioengineering-09-00697-t002:** Fitting results with the proposed electrical circuit (Figure 2) using the Levenberg–Marquardt optimization algorithm.

c_i_ [p/μL]	Diameter	$T_{s}$ × 10^−3^	$P_{s}$	$C_{e}$ [μF]	$R_{d}$ [Ω]	$T_{d}$ × 10^−7^	$P_{d}$
c_1_	$\emptyset_{1}$	0.86	0.31	0.57	3815	9.9	0.73
	$\emptyset_{2}$	1.86	0.24	0.74	6412	7.1	0.74
	$\emptyset_{3}$	0.62	0.36	0.84	7284	6.9	0.75
c_2_	$\emptyset_{1}$	1.54	0.26	0.62	3868	9.4	0.75
	$\emptyset_{2}$	1.43	0.26	0.82	3570	6.9	0.71
	$\emptyset_{3}$	0.54	0.37	0.85	6753	6.7	0.75
c_3_	$\emptyset_{1}$	0.89	0.31	0.66	3863	9.3	0.76
	$\emptyset_{2}$	8.50	0.44	0.72	1186	8.6	0.8
	$\emptyset_{3}$	1.05	0.3	0.86	7078	7.2	0.75
c_4_	$\emptyset_{1}$	1.44	0.27	0.71	4561	8.2	0.78
	$\emptyset_{2}$	5.18	0.15	0.76	1741	8.3	0.8
	$\emptyset_{3}$	0.16	0.29	0.88	7875	7.7	0.76
c_5_	$\emptyset_{1}$	3.32	0.19	0.76	5049	8	0.75
	$\emptyset_{2}$	6.63	0.12	0.77	1827	8	0.81
	$\emptyset_{3}$	0.97	0.28	0.90	10692	6.6	0.74

**Table 3 bioengineering-09-00697-t003:** Volume fraction for suspensions with particles of $\emptyset_{1}$, $\emptyset_{2}$ and $\emptyset_{3}$.

		V_f_ × 10^−4^	
Sample	$\emptyset_{1}$	$\emptyset_{2}$	$\emptyset_{3}$
c_1_	0.03	0.4	14
c_2_	0.06	0.9	29
c_3_	0.1	2	58
c_4_	0.2	4	120
c_5_	0.5	7	230

## Data Availability

The data presented in this study are available on request to the corresponding author.

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
