# Peer review of "Effective Capacitance from Equivalent Electrical Circuit as a Tool for Monitoring Non-Adherent Cell Suspensions at Low Frequencies"

_bioengineering, 2022, doi:10.3390/bioengineering9110697_

Round 1
Reviewer 1 Report
L21, p1: We can’t say that “examine polymer particles of 6, 15, and 48 μm in diameter suspended in a 0.9% sodium chloride solution, which mimics biological cells” because these particles should be also charged to do so. There are a lot of papers on that kind of study. Please add the refs.
L29, p1 / L160, p4 and L169, p5, etc: The notion of Cef is definitively not new, more references should be added. It can’t be a strong result of this study because the link between Cef and the concentration is already known.
L13, p5: this section is too short. The method should be more precisely described, especially concerning the time and temperature parameters and on the geometry of the sensor (size, volume, etc).
L176, p5: the choice of the particle sizes is not clearly explained.
L208, p6: in such approach, it seems crucial to precise the time when the measurement is done after having put the suspension in contact with the sensor, because the medium itself is subjected to strong modifications due to the metabolites and because the sweep of the frequency is taking some time, of course. tHis is very important to emphasize this point because it is a main result in the conclusion.
L217, p6: the numerical simulation is not well presented and the link with the rest of the study is not sufficiently explained
Unclear sentences due to English mistakes:
- L26, p1 / L345, p11: Be careful with the math notations (example : 0.03*10-2% to 58*10-2% seems not correct)
- L38, p1: “The electrical double layer (EDL) is a phenomenon that occurs at low frequencies generated at α dispersion” should be rewritten in a clearer way (alpha dispersion is more a consequence)
- L55 p2: is the Gouy-Chapman-Stern model
Reviewer 2 Report
The paper presents a study on the use of Electrical Impedance Spectroscopy (EIS) for the evaluation of biological cell concentration. Measurements have been carried out at low frequencies (10Hz – 1kHz) using both PMMA particles and real cells (K-562 cells and leukocytes). The paper is interesting. I suggest the following revisions:
1) The experimental results have shown how the electrical parameters are function of both the cell concentration and its size. Thus, it is not possible to estimate the cell concentration if the cell size is not known in advance. Have the authors investigated a technique to estimate both the cell concentration and size (for example with measurements on a larger frequency range)?
2) At lines 179-180 there is the sentence: “Resulting five concentrations (ci for i=1,2,3,4,5) of 400, 200, 100, 50, and 25 p/μL for each particle size”. However, from Figure 4 it seems that c1 is the lowest concentration and c5 the highest. If this is the case I suggest to change the sentence to “Resulting five concentrations (ci for i=1,2,3,4,5) of 25, 50, 100, 200, and 400 p/μL for each particle size”.
3) At lines 326-328 there is the sentence: “The goodness of the fit for K-562 cells is R2=0.95 with a sensitivity of 0.07 μL*F/cells, showing a clear relationship with cell concentration. For leukocytes, R2=0.95 with a sensitivity of 0.28 μL*F/cells”. However, in Fig. 7 (b) the coefficient of determination for leukocytes is 0.92 and not 0.95.
